# MKP-1 modulates ubiquitination/phosphorylation of TLR signaling

Jaya Talreja[1], Christian Bauerfeld[2], Xiantao Wang[3], Markus Hafner[3], Yusen Liu[4] , Lobelia Samavati[1,5]

Ubiquitination and phosphorylation are reversible posttranslational protein modifications regulating physiological and pathological processes. MAPK phosphatase (MKP)-1 regulates innate and adaptive immunity. The multifaceted roles of MKP-1 were attributed to dephosphorylation of p38 and JNK MAPKs. We show that the lack of MKP-1 modulates the landscape of ubiquitin ligases and deubiquitinase enzymes (DUBs). MKP-1$^{-/-}$ showed an aberrant regulation of several DUBs and increased expression of proteins and genes involved in IL-1/TLR signaling upstream of MAPK, including IL-1R1, IRAK1, TRAF6, phosphorylated TAK1, and an increased K63 polyubiquitination on TRAF6. Increased K63 polyubiquitination on TRAF6 was associated with an enhanced phosphorylated form of A20. Among abundant DUBs, ubiquitin-specific protease-13 (USP13), which cleaves polyubiquitin-chains on client proteins, was substantially enhanced in murine MKP-1–deficient BMDMs. An inhibitor of USP13 decreased the K63 polyubiquitination on TRAF6, TAK1 phosphorylation, IL-1$\beta$, and TNF-$\alpha$ induction in response to LPS in BMDMs. Our data show for the first time that MKP-1 modulates the ligase activity of TRAF6 through modulation of specific DUBs.

## Introduction

MAPK phosphatase (MKP)-1 dephosphorylates TXY motifs on MAPKs, thereby negatively regulating MAPKs that are involved in the synthesis of pro-inflammatory cytokines (Liu et al, 2007; Wang et al, 2007; Wang et al, 2008). MKP-1 regulates the activities of numerous transcription factors of inflammatory genes (Liu et al, 2008; Talwar, Bauerfeld et al, 2017a; Bauerfeld et al, 2020). MKP-1 plays a significant role in the pathogenesis of inflammation and metabolic diseases including sepsis, asthma, sarcoidosis, obesity, and type II diabetes (Zhao et al, 2006; Rastogi et al, 2011; Lawan et al, 2018). Recently, we have shown that MKP-1–deficient BMDMs exhibit marked up-regulation of key mitochondrial proteins involved in oxidative phosphorylation (Bauerfeld et al, 2012; Bauerfeld et al,

2020). It is widely accepted that MKP-1 preferentially dephosphorylates p38 and JNK, but it can also act on ERK (Zhao et al, 2005; Zhao et al, 2006; Wang et al, 2007, 2008). The multitude effects of MKP-1 on the innate immunity, adaptive immunity, cellular metabolism, and in cancer biology raise an interesting question of whether all these effects are dependent on MAPKs deactivation.

The innate immune response is activated by pathogen-associated molecular patterns through a family of TLRs (Medzhitov & Janeway, 2000). TLR signaling can be classified into MyD88-dependent or MyD88-independent pathways. In the MyD88-dependent pathway, after detecting pathogen-associated molecular patterns, MyD88 is recruited to TLRs with interleukin 1 receptor–associated kinases (IRAKs) and activates a ubiquitin E3 ligase, the TNF receptor–associated factor 6 (TRAF6) (Metzger et al, 2014). It has been proposed that TRAF6 functions as docking site for formation of signaling complexes, and that K63-linked autoubiquitination of TRAF6 serves as scaffold to recruit Transforming growth factor $\beta$-activated kinase (TAK)1 to activate multiple downstream signaling pathways (Walsh et al, 2008; Walsh et al, 2015). TLR signaling is tightly regulated to maintain immune homeostasis, and both hyperactivation and hypoactivation of TLR signaling can cause human diseases (Opipari et al, 1990). Reversible phosphorylation/dephosphorylation and ubiquitination/deubiquitination of the pathway mediators assures cellular homeostasis in response to pathogens. TRAF6 plays a vital role in signal transduction both in innate and adaptive immunity by bridging signaling from TNFR, TLR/IL-1R, TCR, IL-17R, and B-cell receptor (Wu & Arron, 2003; Walsh et al, 2008).

Ubiquitination occurs in a three-step reaction mediated by three different enzymes: an ubiquitin-activating enzyme (E1), an ubiquitin conjugating enzyme (E2), and an ubiquitin ligase enzymes (E3). Ubiquitin is first activated by E1, followed by conjugation to an E2 before being finally ligated to the lysine residues of target proteins by the E3 ligase (Metzger et al, 2014). The E3 ligase activity and its ability to recognize targeted proteins is regulated through post-translational modification. Deubiquitinating enzymes (DUBs) oppose the function of E3 ligases by cleaving ubiquitin chains (Komander et al, 2009; He et al, 2016). There are more than 500 genes encoding DUBs in the human genome. There are six families of DUBs: ubiquitin-specific proteases (USPs), ubiquitin carboxy-terminal hydrolases (UCHs), ovarian-tumor proteases (OTUs),

---

[1]Department of Medicine, Division of Pulmonary, Critical Care and Sleep Medicine, Wayne State University School of Medicine and Detroit Medical Center, Detroit, MI, USA [2]Department of Pediatrics, Division of Critical Care, Central Michigan University, Mount Pleasant, MI, USA [3]Laboratory of Muscle Stem Cells and Gene Regulation, National Institute of Arthritis and Musculoskeletal and Skin Disease, National Institutes of Health, Bethesda, MD, USA [4]Center for Perinatal Research, The Abigail Wexner Research Institute at Nationwide Children's Hospital, Columbus, OH, USA [5]Center for Molecular Medicine and Genetics, Wayne State University School of Medicine, Detroit, MI, USA

Correspondence: ay6003@wayne.edu

Machado–Joseph disease protein domain proteases (MJDs), JAMM/MPN domain-associated metallopeptidases (JAMMs), and the recently discovered monocyte chemotactic protein-induced protein (MCPIP) family (Reyes-Turcu et al, 2009). A20 (encoded by *TNFAIP3* gene) is known as an editing enzyme with ability to act as ligase and DUB (Coornaert et al, 2008). A20 is an inducible and broadly expressed cytoplasmic protein that inhibits TRAF6-induced NF-κB activity (Coornaert et al, 2008; Kondo et al, 2012). A20 deubiquitylates TRAF6 to tune down TRAF6-mediated signaling (Heyninck & Beyaert, 1999; Coornaert et al, 2008; Kondo et al, 2012).

In this study, we show that the expression of several TLR signaling molecules, including IL-1R1, IRAK1, and TRAF6, are significantly increased in MKP-1$^{-/-}$ deficient mice and BMDMs at baseline and in response to sepsis and LPS stimulation, respectively. Because TRAF6 regulates innate, adaptive immunity and metabolism, we hypothesize that MKP-1 modulates TRAF6 activity through regulating its ubiquitination, thus enhancing the downstream cascades. Increased TRAF6 K63 ubiquitination was associated with abundance of A20 protein along with its phosphorylated form. In addition, MKP-1 deficiency led to up-regulation of several DUBs including USP13. We show that USP13 modulates TRAF6 and TLR signaling and inhibition of USP13 attenuates IL-1β production. Our studies suggest that in addition to inhibiting the MAPK pathways, MKP-1 may also modulate the ubiquitination process of critical signaling regulators in the proximity of the TLRs.

## Results

### Differential expression of TLR signaling molecules and deubiquitinases in wild-type (WT) and MKP-1$^{-/-}$ mice

Given the fact that TLR signaling regulates inflammatory signaling in sepsis, we hypothesize that MKP-1 deficiency leads to an aberrant TLR signaling and the observed hyperinflammation in these mice in response to sepsis. To advance our understanding of the role of MKP-1 deficiency on TLR signaling, we performed data mining of our previously published RNA sequencing results (Li et al, 2018; Bauerfeld et al, 2020). First, we compared differentially expressed (DE) genes between WT and MKP-1$^{-/-}$ mice at baseline and in response to *Escherichia coli* infection using a DE-seq program. Furthermore, we performed pathway analysis using Gene Trail 2 analysis on the DE genes. The TLR pathway was highly impacted in MKP-1–deficient mice. Fig 1A shows an expression heat map of the genes involved in IL-1R/TLR signaling in WT and MKP-1$^{-/-}$ mice in response to *E. coli* sepsis. We found an increased expression of several important regulators of TLR signaling, including IL-1R1, TRAF6, MAP3K6. The second highly impacted pathway was the ubiquitination pathway. We focused on established negative regulators of the ubiquitination process, especially the DUBs. Fig 1B shows the mRNA expression heat maps of several DUB genes. A number of DUB genes were up-regulated in MKP-1$^{-/-}$ mice compared to WT mice after *E. coli* infection, including TNFAIP3 (also known as A20), USP 13, USP17ld, and USP18. Induction of USP43 after *E. coli* infection was attenuated in MKP-1$^{-/-}$ mice. As shown, mRNA expression for IL-1R1 (Fig 1C) and TRAF6 (Fig 1D) is markedly

increased in sepsis, especially in MKP-1$^{-/-}$ mice. Fig 1E and F depict the quantitative changes in the expression of TNFAIP3 and USP13. These data suggest that sepsis has a significant effect on the expression of DUBs, including A20 and USP13 in vivo in both WT and MKP-1$^{-/-}$ mice. In contrast, as Fig S1A–C shows that there were no significant differences in the expression of IRAK4 (A), TRIF (B) and MyD88 (C) genes at baseline or after *E. coli* infection between WT and MKP-1$^{-/-}$ mice. Similarly, Fig S1D and E shows that the expression of CYLD (D) and USP21 (E) were not significantly affected. In contrast, we found that expression of genes for pro-inflammatory cytokines including, IL-1β, IL-6 and TNF-α, markedly increased after *E. coli* infection in MKP-1$^{-/-}$ mice Fig S1F–H. To validate the RNA-seq data, we determined the expression levels of the selected genes in BMDMs. BMDMs from WT and MKP-1$^{-/-}$ mice were challenged with LPS for 1 h, RNA was isolated and subjected to qRT-PCR using selected primers. MKP-1$^{-/-}$ BMDMs exhibit significantly higher expression of IL-1R1 at baseline as compared to WT BMDMs with no further increase in response to LPS (Fig 1G). TRAF6 expression was significantly increased in LPS-stimulated WT and MKP-1$^{-/-}$ BMDMs (Fig 1H). TNFAIP3 (A20) gene expression was slightly higher in MKP-1$^{-/-}$ BMDMs, but LPS stimulation led to an enhanced TNFAIP3 in both WT and MKP-1$^{-/-}$ BMDMs (Fig 1I). MKP-1$^{-/-}$ BMDMs exhibited significantly higher expression of USP13 at baseline as compared to WT BMDMs with no further increase in response to LPS challenge. The expression of USP13 increased significantly in LPS-treated WT BMDMs (Fig 1J). There were differences in BMDMs and livers in gene expression profiles, but similarities in elevated basal levels of IL-1R, TRAF6, TNFAIP3, and USP13 in MKP-1. These raise the intriguing possibility that MKP-1 deficiency modulates a gene regulatory network in the proximity of the TLR signaling.

### MKP-1–deficient BMDMs exhibit marked changes at the level of signaling molecules in the proximity of the IL-1 receptor

Rapid phosphorylation/dephosphorylation and ubiquitination/deubiquitination processes regulate activation of TLR pathway. To gain mechanistic insights at the protein level beyond gene expression data set, we investigated the effect of MKP-1 deficiency on the expression of the upstream TLR/IL-1R signaling in BMDMs derived from WT and MKP-1$^{-/-}$ mice. BMDMs were challenged with LPS (100 ng/ml) for various time periods. Whole cell lysates were immunoblotted with specific antibodies against IL-1R and β-actin (loading control) (Fig 2A). The mean densitometric analysis of three independent experiments show that IL-1R1 is highly abundant in MKP-1$^{-/-}$ BMDMs both at baseline with minimal increment after LPS challenge compared with WT BMDMs (Fig 2B). Next, we determined the expression of IRAK1. The results indicate that MKP-1$^{-/-}$ BMDMs exhibit significantly higher IRAK1 expression at baseline than WT BMDMs (Fig 2C and D). Consistent with previous published results (Kollewe et al, 2004), upon activation IRAK1 levels decreased in WT and MKP-1$^{-/-}$ BMDMs. Furthermore, we determined the expression levels of IRAK4 protein. Similar to IRAK1 levels, MKP-1$^{-/-}$ BMDMs exhibit significantly an abundance of IRAK4 protein at baseline (Fig 2C). These data indicate that many upstream TLR signaling proteins are increased in MKP-1$^{-/-}$ BMDMs. As TLR/IL-1R activation induces the phosphorylation of IRAK1, we assessed the phosphorylated (p) IRAK1. MKP-1$^{-/-}$ BMDMs exhibit significantly higher phosphorylated

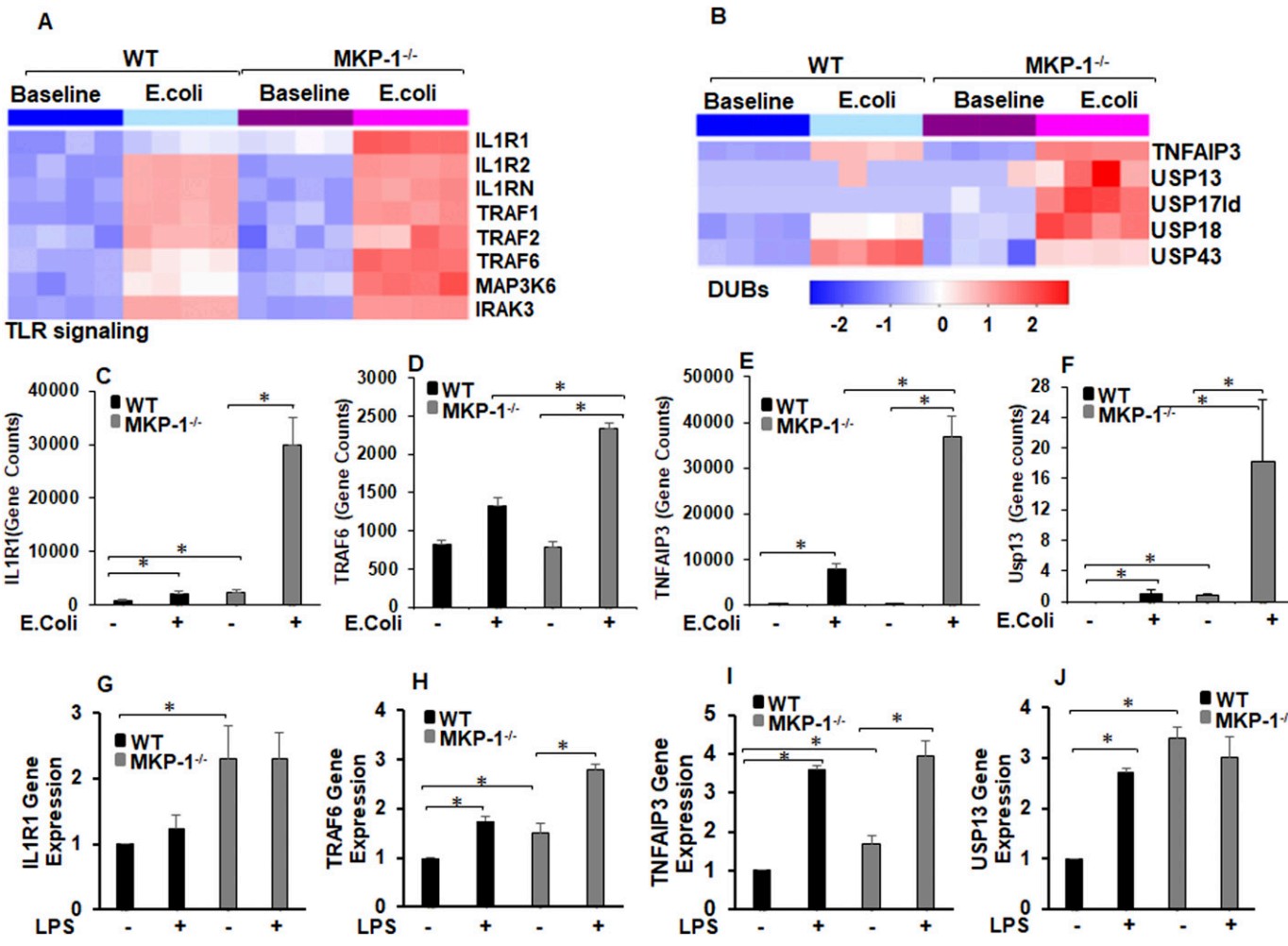

**Figure 1. Expression of genes related to IL-1R/TLR signaling and ubiquitin-editing enzymes (DUBs) altered by *Escherichia coli* infection in WT and MKP-1 knockout mice.**

**(A)** Heat map showing relative changes in expression of select transcripts related to IL-1R/TLR signaling in livers of mice using RNA-seq. The color gradient ranges from red (highest level of expression) to blue (lowest level of expression), with light blue or white representing intermediate levels. Note: Each lane represents a different animal, only significantly affected genes are shown ($\log_2$-fold change > 0.5 and FDR < 5%, n = 4). **(B)** Heat map showing relative changes in expression of select transcripts related to DUBs in livers of mice using RNA-seq. The color gradient ranges from red (highest level of expression) to blue (lowest level of expression), with light blue or white representing intermediate levels. Note: Each lane represents a different animal, only significantly affected genes are shown in the heat map ($\log_2$-fold change > 0.5 and FDR < 5%, n = 4). **(C, D, E, F)** Normalized transcript counts of (C) IL-1R1, (D) TRAF6, (E) TNFAIP3, and (F) USP13 with four replicates per condition. * represents a *P*-value < 0.05 (*t* test). **(G, H, I, J)** BMDMs from WT and MKP-1$^{-/-}$ mice were cultured and challenged with LPS (100 ng/ml) for 1 h. Total RNA was extracted from the cells (n = 4). Isolated RNA was reverse-transcribed by the reverse transcription system using iQSYBR Green Supermix. Relative mRNA levels were calculated by normalizing to GAPDH. **(G, H, I, J)** The relative expression fold change of genes (G) IL-1R1, (H) TRAF6, (I) TNFAIP3, and (J) USP13. Data were analyzed using the paired, two-tailed *t* test, and the results were expressed as fold change. * represents a *P*-value < 0.05.

form of IRAK1 even without LPS stimulation (Fig 2E and F). WT and MKP-1$^{-/-}$ BMDMs responded to LPS challenge (15 min) with increased phosphorylation of IRAK1, which decreased to baseline after 30 min. TRAF6 mediates the signaling from members of both the TLR/IL-1 and the TNF receptor family. Upon ligand binding on IL-1/TLR receptors TRAF6 is auto-ubiquitinated on lysine 63 (K63), which enhances its ligase function to activate TLR signaling molecules (Walsh et al, 2015; Cohen & Strickson, 2017). TRAF6 interacts with upstream kinases such as IRAKs and downstream kinases, including TAK1 and IκB kinases to activate the MAPK pathway (Dong et al, 2006; Wertz & Dixit, 2010; Walsh et al, 2015). Because of the increased expression of IL-1R1 and IRAK1 in MKP-1–deficient BMDMs (Fig 2A and C), we examined the effect of MKP-1 deficiency on the

expression of TRAF6 in these cells. BMDMs derived from WT and MKP-1$^{-/-}$ mice were cultured in the absence or presence of LPS and whole cell lysates were immunoblotted with specific antibodies against TRAF6 and β-actin. We found that MKP-1$^{-/-}$ BMDMs expressed significantly higher levels of TRAF6 at baseline and after LPS stimulation than did WT BMDMs (Fig 2G and H). Previously, our groups and others have shown that MKP-1 deficiency leads to enhanced phosphorylation of p38 and JNK in response to TLR activation (Zhao et al, 2005, 2006; Talwar et al, 2017a; Bauerfeld et al, 2020). Because it is possible that there is a positive feedback mechanism by the p38 and JNK regulating TRAF6 expression, we determined if inhibition of p38 and JNK modulates TRAF6 expression in MKP-1 deficient BMDMs. As shown in Fig S2A, neither

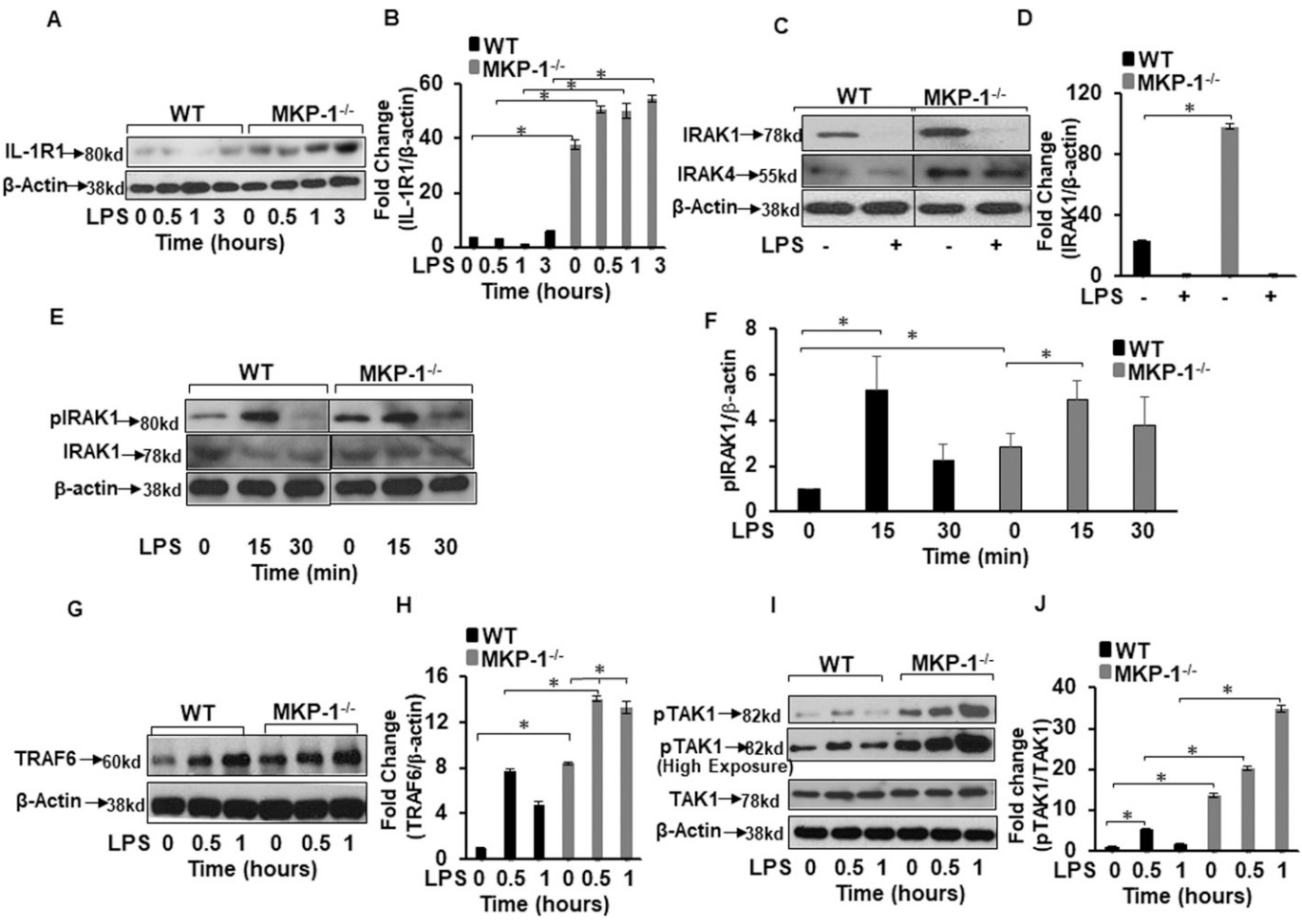

**Figure 2. MKP-1$^{-/-}$ BMDMs exhibit higher IL-1R1, IRAK1, IRAK4, pIRAK1, TRAF6, and pTAK1.**
BMDMs from WT and MKP-1$^{-/-}$ mice were cultured and challenged with LPS (100 ng/ml) for 0.5, 1, and 3 h. Whole cell extracts were prepared and subjected to SDS–PAGE and Western blot analysis using specific antibodies against IL-1R1, IRAK1, pIRAK1, TRAF6, and pTAK1. Equal loading was confirmed using an antibody against β-actin. **(A)** MKP-1$^{-/-}$ BMDMs expressed substantially elevated IL-1R1 relative to WT cells. **(B)** Densitometry values expressed as fold increase of the ratio of IL-1R1/β-actin. **(C)** MKP-1$^{-/-}$ BMDMs exhibited higher IRAK1 and IRAK4 expression at baseline as compared to WT. **(D)** Densitometry values expressed as fold increase of the ratio of IRAK1/β-actin. **(E)** MKP-1$^{-/-}$ BMDMs exhibited higher pIRAK1 level at baseline as compared to WT, which increased after 15 min of activation with LPS in both WT and MKP-1$^{-/-}$ BMDMs. **(F)** Densitometry values expressed as fold increase of the ratio of p-IRAK1/β-actin. MKP-1$^{-/-}$. **(G)** BMDMs constitutively expressed significantly higher TRAF6 as compared to WT. **(H)** Densitometry values expressed as fold increase of the ratio of TRAF6/β-actin. **(I)** MKP-1$^{-/-}$ BMDMs exhibited higher pTAK1 expression at baseline as compared to WT. **(J)** Densitometry values expressed as fold increase of the ratio of pTAK1/TAK1. All densitometry data represent mean ± SEM of at least three independent experiments. * represents a $P$-value < 0.05.
Source data are available for this figure.

specific inhibitor of p38 (SB SB203580) nor JNK (SP600125) decreased the expression of TRAF6 mRNA levels. Similarly, the effect of both inhibitors on TRAF6 protein levels was not significant (Fig S2B and C). These data indicate that TRAF6 expression is not regulated through activation of p38 and JNK in MKP-1–deficient cells. K63-linked poly-ubiquitination on TRAF6 activates TAK1 (Irie et al, 2000; Talreja & Samavati, 2018). We then asked if the increased TRAF6 protein in MKP-1$^{-/-}$ BMDMs translates into an increase in TAK1 phosphory-lation. As compared with WT BMDMs, MKP-1$^{-/-}$ BMDMs constitu-tively express higher levels of phosphorylated TAK1 (pTAK1) (Fig 2I). Moreover, LPS challenge led to further increased and prolonged TAK1 phosphorylation in MKP-1$^{-/-}$ BMDMs (Fig 2I and J). Taken together, our results clearly show that MKP-1 deficiency considerably alters the signaling machinery in the proximity of the TLR/IL-1 receptor.

**Increased K63-linked TRAF6 polyubiquitination and up-regulation of ubiquitin conjugation enzyme 13 (Ubc13) in MKP-1–deficient BMDMs**

Whereas K63-linked polyubiquitination on TRAF6 plays a key role in mediating the activation signal during inflammatory response, the K48-linked ubiquitination of proteins targets the substrates for proteasome degradation and termination of the inflammatory signaling. Because MKP-1$^{-/-}$ BMDMs exhibited significantly higher TRAF6 levels and greater pTAK1 than WT BMDMs, we asked whether MKP-1 deficiency alters the balance of the two types (K48 versus K63) of polyubiquitination on TRAF6. BMDMs derived from WT and MKP-1$^{-/-}$ mice were stimulated with LPS for 30 min or left untreated. Whole cell lysates were immunoprecipitated (IP) using a TRAF6

antibody, and the immunoprecipitates were immunoblotted using an anti-K63 polyubiquitin antibody. As shown in Fig 3A, MKP-1$^{-/-}$ BMDMs exhibit a significantly higher level of K63-linked poly-ubiquitinated TRAF6 as compared with WT BMDMs both at baseline and after LPS stimulation. We repeated the same experiments by performing TRAF6 immunoprecipitation followed by immunoblot-ting using antibody against K48 polyubiquitin (Fig 3B). In response to LPS challenge, the levels of K48-linked polyubiquitinated TRAF6 were increased in WT BMDMs, but not in MKP-1$^{-/-}$ BMDMs (Fig 3B). These results suggest that MKP-1$^{-/-}$ BMDMs predominantly undergo K63-linked polyubiquitination on TRAF6, leading to a prolonged TRAF6 and TAK1 activation. In contrast, whereas K63-linked TRAF6 polyubiquitination also occurs in WT BMDMs, the rapidly enhanced K48-linked polyubiquitination of TRAF6 dampens TRAF6 signaling after LPS challenge in these cells, thus restraining the propagation of the inflammatory signaling. TRAF6 functions as a signaling platform for MyD88 and IRAKs, resulting in activation of the TAK1/TAB1/TAB2 complexes (Wertz & Dixit, 2010). Therefore, we deter-mined if there is an increased association of IRAK1 with TRAF6 in MKP-1$^{-/-}$ BMDMs. The immunoblot of immunoprecipitated TRAF6 using antibody against IRAK1 revealed higher IRAK1 levels in MKP-1$^{-/-}$ BMDMs as compared with levels in WT BMDMs at baseline and in response to LPS challenge (Fig 3C).

Ubiquitin conjugating enzyme (Ubc) 13 is an E2 enzyme that forms K63-linked ubiquitin chains specifically for its cognate E3 ligases (Fukushima et al, 2007; Petroski et al, 2007). Ubc13 forms a complex with TRAF6 (E3 ligase) to promote K63-linked ubiq-uitination on selected substrates (Fukushima et al, 2007; Petroski et al, 2007). Ubc13 is critical for ubiquitination and activation of TRAF6. Because of increased expression of IL-1R1 and IRAK1, altered TRAF6 ubiquitination and signaling complex formation in MKP-1–deficient BMDMs, we examined if MKP-1 deficiency is associated with an aberrant Ubc13 expression. BMDMs derived from WT and MKP-1$^{-/-}$ mice were cultured in the absence or presence of LPS and whole cell lysates were immunoblotted with specific antibodies against Ubc13. Ubc13 expression was significantly higher in MKP-1–deficient BMDMs at baseline with no further increase in response to LPS challenge (Fig 3D and E). In WT BMDMs, LPS stimulation resulted in a con-siderable increase in Ubc13 levels within 3 h, this remained elevated for at least 24 h. Higher basal level of Ubc13 in MKP-1$^{-/-}$

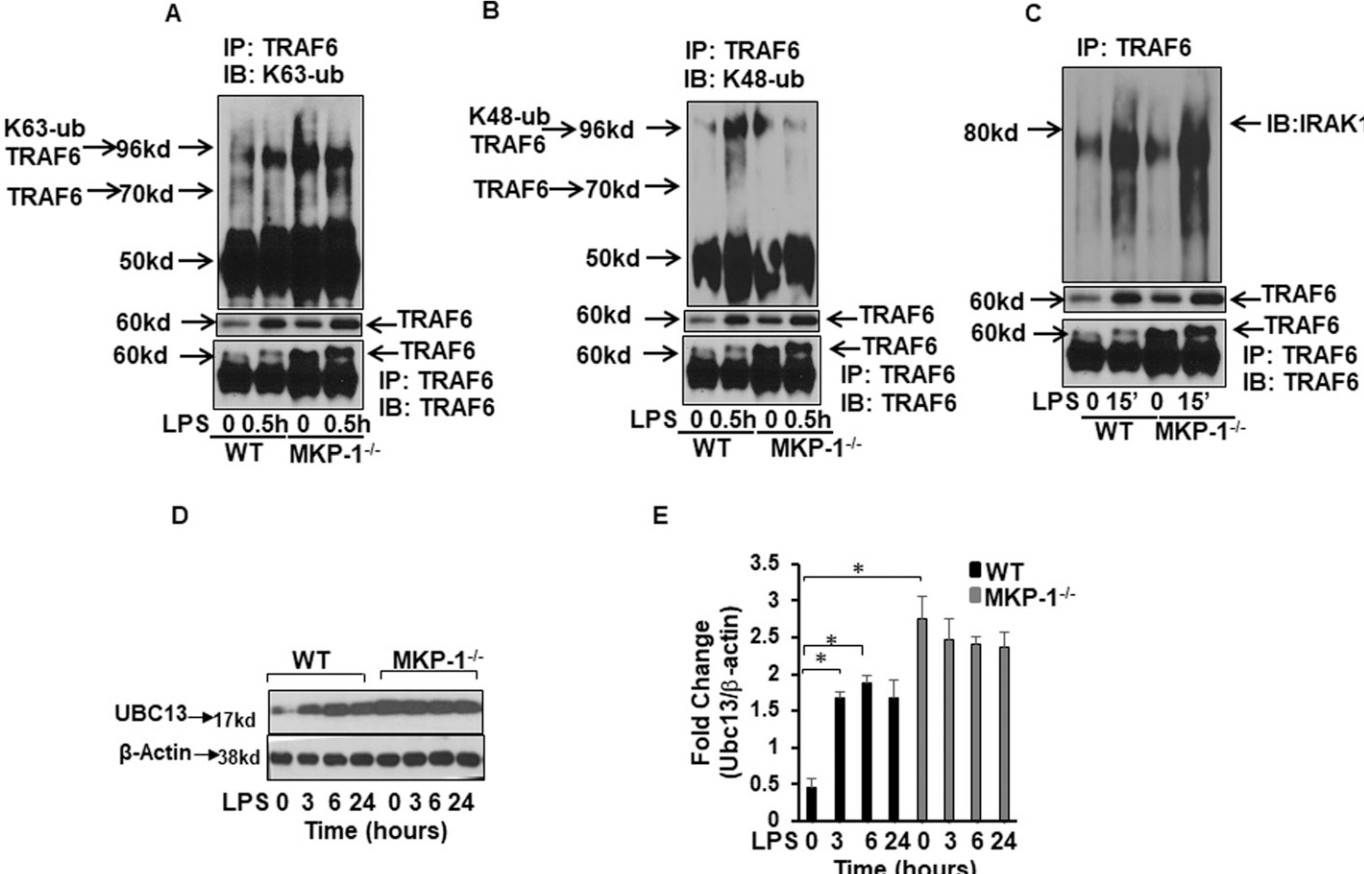

**Figure 3. Differential ubiquitination of TRAF6 in MKP-1$^{-/-}$ BMDMs as compared with WT BMDMs.**
WT or MKP-1$^{-/-}$ BMDMs were cultured in the presence or absence of LPS for 30 min. Protein lysates were immunoprecipitated with TRAF6-specific antibody and equal amounts of immunoprecipitates were subjected to SDS–PAGE. Western blot analysis was performed using specific antibodies against K63- or K48-ubiquitin. **(A)** MKP-1$^{-/-}$ BMDMs exhibit higher K63 protein in immunoprecipitated TRAF6 as compared with WT BMDMs. The lower panel shows the expression of TRAF6 in the cell lysates used for immunoprecipitation. **(B)** WT BMDMs exhibit higher K48 protein in immunoprecipitated TRAF6. **(C)** MKP-1$^{-/-}$ BMDMs exhibit higher expression of IRAK1 in immunoprecipitated TRAF6. **(D)** MKP-1$^{-/-}$ BMDMs exhibited higher Ubc13 expression at baseline as compared to WT. **(E)** Densitometry values expressed as fold increase of the ratio of Ubc13/$\beta$-actin. All densitometry data represent mean ± SEM of at least three independent experiments. * represents a $P$-value < 0.05.

macrophages indicates that MKP-deficiency results in an increased generation of K63-linked ubiquitin chains for the increased expression of ubiquitination enzymes involved in TLR/IL-1R signaling.

### MKP-1 deficiency alters A20 expression and phosphorylation

A20 is an ubiquitin-editing enzyme and the best-known negative regulator of TRAF6 signaling (Vereecke et al, 2009; Matmati et al, 2011). A20 cleaves K63-linked polyubiquitin chains from selective substrates (e.g., TRAF6) and builds degradative K48-linked ubiquitin chains on TRAF6. A20 disrupts E2-E3 ubiquitin ligase interactions by destabilizing ubiquitin-enzyme complexes (Shembade et al, 2010). Both mechanisms lead to the termination of TLR/TRAF6 activation signals (Kondo et al, 2012). Thus, we determined the protein level of A20 in WT and MKP-1$^{-/-}$ BMDMs. Unstimulated WT BMDMs exhibited low abundance of A20, the expression increased 3 h post LPS challenge (Fig 4A). Surprisingly, MKP-1$^{-/-}$ BMDMs expressed significantly higher A20 levels at baseline and its levels first decreased in response to LPS but rapidly increased after 1 h in MKP-1$^{-/-}$ BMDMs (Fig 4A). The mean densitometric analysis of three independent experiments is shown in Fig 4B. These data indicate that enhanced K63-linked ubiquitination of TRAF6 is not due to a lack of A20 protein in MKP-1 –deficient BMDMs. A20 function is regulated partly by phosphorylation (Wertz et al, 2004). As MKP-1 is a phosphatase, we asked if MKP-1 deficiency leads to an aberrant A20 phosphorylation. BMDMs derived from WT and MKP-1$^{-/-}$ mice were cultured in the absence or presence of LPS and whole cell lysates were immunoblotted using specific antibody against phospho-A20 (Fig 4C). Equal loading was confirmed by immunoblotting using $\beta$-actin antibody. The mean densitometric analysis of three independent experiments is shown in Fig 4D. WT BMDMs exhibited lower baseline levels of phosphorylated form of A20, which increased upon LPS challenge (Fig 4C). In contrast, MKP-1$^{-/-}$ BMDMs expressed significantly higher baseline phosphorylated form of

A20, which increased further in response to LPS challenge. The pA20 levels peaked in both cell types at 3–6 h post LPS treatment (Fig 4C and D).

### DUB inhibitors decrease the expression of TRAF6, pTAK1, and modulate TRAF6 activity through K63/K48-linked polyubiquitination on TRAF6

To explore the role of DUBs on TRAF6 and its downstream target TAK1 during the inflammatory response in macrophages, we took advantage of two different DUB inhibitors: an universal DUB inhibitor (PR-619, designated as DUBi) and another inhibitor with predominant activity against USP13 but also activity against USP10 (spautin-1, designated as Usp13i) (Liu et al, 2011; Seiberlich et al, 2013). BMDMs were treated with either DUBi or the more specific Usp13i for 30 min, and then activated with LPS for 3 h. Whole cell lysates were immunoblotted using specific antibodies against TRAF6, phospho-TAK1, and TAK1 (Fig 5A). Equal loading was confirmed by immunoblotting using $\beta$-actin antibody. The mean densitometric analyses of three independent experiments are shown for TRAF6 and pTAK1 (Fig 5B and C, respectively). Interestingly, neither DUBi nor Usp13i decreased the expression of LPS-induced TRAF6 levels in WT BMDMs. In fact, treatment of WT BMDMs with both inhibitors enhanced the TRAF6 levels. In contrast, DUBi and Usp13i, significantly decreased baseline and LPS-induced TRAF6 expression in MKP-1$^{-/-}$ BMDMs (Fig 5A and B). Similarly, neither DUBi nor Usp13i decreased the pTAK1 levels in WT BMDMs. However, DUBi and Usp13i significantly decreased pTAK1 at baseline and in response to LPS in MKP-1$^{-/-}$ BMDMs (Fig 5A and C). Because both, the non-selective DUBi and Usp13i, decreased baseline and LPS-induced TRAF6 levels in MKP-1$^{-/-}$ BMDMs, we determined the effect of Usp13i and DUBi on the level of K63 and K48-linked polyubiquitination on TRAF6. MKP-1$^{-/-}$ BMDMs were treated with Usp13i or DUBi for 30 min and then challenged with LPS for 3 h. TRAF6 in cell lysates was

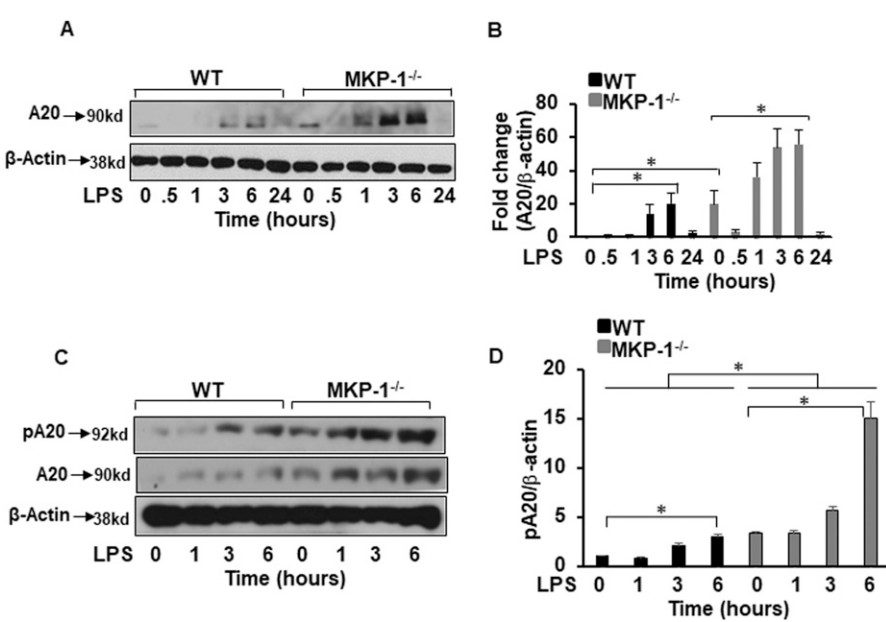

**Figure 4. Differential expression of A20 and phosphorylated A20 in MKP-1$^{-/-}$ BMDMs as compared with WT BMDMs.**
**(A)** MKP-1$^{-/-}$ BMDMs exhibited higher A20 expression at baseline with a significant increase after LPS stimulation as compared with WT. **(B)** Densitometry values expressed as fold increase of the ratio of A20/$\beta$-actin. **(C)** MKP-1$^{-/-}$ BMDMs exhibited higher expression of pA20 at baseline as compared with WT BMDMs. Stimulation with LPS for 1, 3 and 6 h increased the expression of pA20 in WT and MKP-1$^{-/-}$ BMDMs. **(D)** Densitometry values expressed as fold increase of the ratio of pA20/$\beta$-actin. All densitometry data represent mean ± SEM of at least three independent experiments. * represents a *P*-value < 0.05.

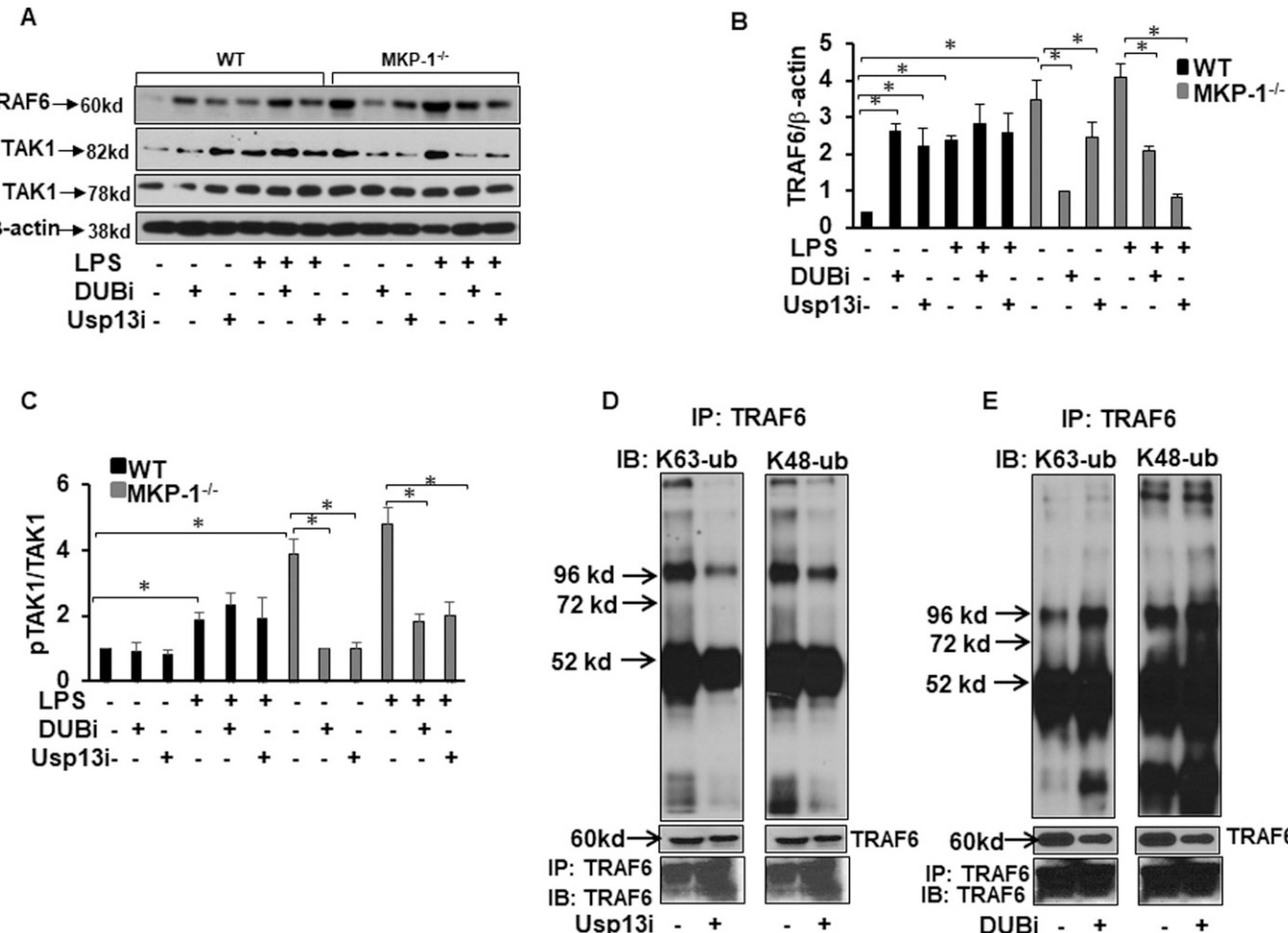

**Figure 5. DUBi and Usp13i inhibit TRAF6, pTAK1 expression and modifies ubiquitination of TRAF6 in MKP-1⁻/⁻ BMDMs.**
BMDMs from WT and MKP-1⁻/⁻ mice were treated with either DUBi (5 µM) or Usp13i (5 µM) for 30 min and then activated with LPS (100 ng/ml) for 3 h. Whole cell extracts were prepared and subjected to SDS–PAGE and Western blot analysis using specific antibodies against TRAF6 and pTAK1. Equal loading was confirmed using an antibody against β-actin. **(A)** DUBi and Usp13i increased the basal as well as LPS-induced TRAF6 and pTAK1 expression in WT BMDMs whereas the inhibitors decreased the levels of TRAF6 and pTAK1 in MKP-1⁻/⁻ BMDMs **(B)** Densitometry values expressed as fold increase/decrease of the ratio of TRAF6/β-actin. **(C)** Densitometry values expressed as fold increase/decrease of the ratio of pTAK1/β-actin. **(D, E)** MKP-1⁻/⁻ BMDMs were treated with Usp13i or DUBi for 30 min and then activated with LPS for 3 h. Whole cell lysates were immunoprecipitated (IP) using TRAF6 antibody followed by immunoblotting using anti-K63 or anti-K48 polyubiquitin antibodies. **(D)** Usp13i decreased the expression of K63-linked polyubiquitinated TRAF6 and K48-linked polyubiquitinated TRAF6. **(E)** DUBi has no effect on the expression of K63-linked polyubiquitinated TRAF6, but it increased the expression of K48-linked polyubiquitinated TRAF6. The lower panel shows the expression of TRAF6 in the cell lysates used for immunoprecipitation.

immunoprecipitated (IP) using TRAF6 antibody and Western blot analysis on the TRAF6 immune complexes was performed using antibodies against K63-linked or K48-linked polyubiquitin. We found that Usp13i decreased the level of both K63-linked polyubiquitinated and K48-linked polyubiquitinated TRAF6 (Fig 5D). In contrast, DUBi had little effect on K63-linked polyubiquitinated TRAF6, but it substantially increased the level of K48-linked polyubiquitinated TRAF6 (Fig 5E). As K48-linked polyubiquitination on TRAF6 is associated with degradation of TRAF6 and the enhanced K48-linked polyubiquitination provides a plausible explanation for the decreased TRAF6 level after DUBi treatment (Fig 5A). Although both Usp13i and DUBi decreased the TRAF6 expression in MKP-1–deficient BMDMs, they have differential effects on TRAF6 polyubiquitination: DUBi increases the K48 polyubiquitination, but it has little effect on K63 polyubiquitination, whereas Usp13i decreases both the K63 and the K48 polyubiquitination.

**DUB inhibitors attenuate IL-1β and TNF-α production in MKP-1⁻/⁻ BMDMs**

Our previous results demonstrate that MKP-1⁻/⁻ BMDMs exhibit higher production of LPS-induced IL-1β production (Talwar et al, 2017a). Therefore, we examined if DUBi or Usp13i modulates pro-IL-1β and IL-1β production. WT and MKP-1⁻/⁻ BMDMs were pretreated either with DUBi, Usp13i, or vehicle for 30 min, followed by LPS stimulation for 3 h. Whole cell lysates were immunoblotted using antibodies against pro-IL-1β and β-actin (equal loading) (Fig 6A). WT BMDMs expressed modest levels of pro-IL-1β in response to LPS. The pretreatment of WT BMDMs with Usp13i and DUBi decreased the LPS-induced pro-IL-1β expression. Consistent with our previous reports (Talwar et al, 2017a; Talwar, Bauerfeld et al, 2017b), MKP-1⁻/⁻ BMDMs responded to LPS challenge with higher pro-IL-1β

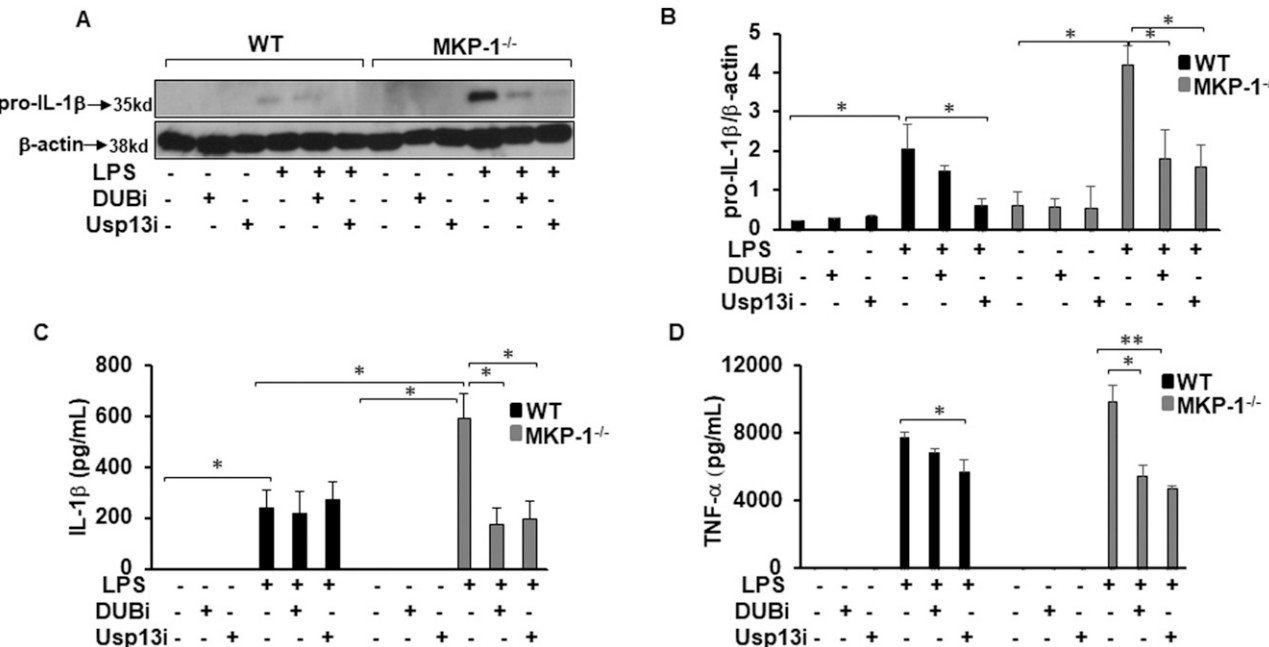

**Figure 6. DUBi and Usp13i inhibit pro-IL-1β expression and IL-1β and TNF-α production in MKP-1⁻/⁻ BMDMs.**
BMDMs from WT and MKP-1⁻/⁻ mice were cultured and challenged with LPS (100 ng/ml) for 3 h in the presence or absence of DUBi (5 μM) or Usp13i (5 μM). Whole cell extracts were prepared and subjected to SDS–PAGE and Western blot analysis using specific antibodies against pro-IL-1β. Equal loading was confirmed using an antibody against β-actin. **(A)** LPS-induced pro-IL-1β expression in both WT and MKP-1⁻/⁻ BMDMs. DUBi and Usp13i inhibited the LPS-induced pro-IL-1β expression in both WT and MKP-1⁻/⁻ BMDMs. **(B)** Densitometry values expressed as fold increase/decrease of the ratio of pro-IL-1β/β-actin. **(C)** DUBi and Usp13i inhibited the LPS-induced IL-1β production in MKP-1⁻/⁻ BMDMs but had no inhibitory effect on IL-1β production in WT BMDMs. **(D)** Usp13i but not DUBi inhibited the LPS-induced TNF-α production in WT BMDMs. Both DUBi and Usp13i inhibited the LPS-induced TNF-α production in MKP-1⁻/⁻ BMDMs. * represents a *P*-value < 0.05 and ** represents a *P*-value < 0.001.

production. DUBi pretreatment modestly inhibited LPS-induced pro-IL-1β expression, whereas USP13i substantially decrease the expression of pro-IL-1β (Fig 6A and B). Likewise, we assessed the effect of both inhibitors on IL-1β production in response to LPS challenge. WT and MKP-1⁻/⁻ BMDMs were pretreated with either DUBi or Usp13i for 30 min, and then stimulated with LPS for 24 h. Released IL-1β levels in the medium were measured by ELISA. Both DUBi and Usp13i significantly decreased the LPS-induced IL-1β production in MKP-1⁻/⁻ BMDMs, but not in WT BMDMs (Fig 6C). Furthermore, we determined the effect of DUBi and Usp13i on TNF-α production (Fig 6D). Usp13i but not DUBi decreased TNF-α production in WT BMDMs. However, both DUBi and Usp13i significantly decreased the LPS-induced TNF-α production in MKP-1⁻/⁻ BMDMs.

## Discussion

Previously, it has been shown that MKP-1 deficiency sensitizes mice to *E. coli* septicemia with enhanced pro-inflammatory cytokine and chemokine production (Frazier et al, 2009). This is generally attributed to prolonged MAPK activation, especially p38 (Zhao et al, 2005, 2006; Talwar et al, 2017a). In addition to regulation of p38 activity, it has been shown that MKP-1 modulates JNK and ERK, as well as JAK/STAT pathways (Huang et al, 2011). Several lines of evidence suggest that MKP-1 modulates various cellular processes beyond MAPK pathways, such as metabolism (Li et al, 2018), mitochondrial

biogenesis (Bauerfeld et al, 2020), p53 (Li et al, 2003), and insulin signaling (Begum et al, 1998). In our present work, we examined the effects of MKP-1 deficiency on the critical signaling molecules in the proximity of TLR signaling and upstream of the MAP kinases, especially on TRAF6/A20 and TAK1. Our studies demonstrate that MKP-1 deficiency markedly alters the levels or activities of upstream TLR signaling molecules, including IL-1R1, IRAK1, TRAF6, and TAK1 (Fig 2). We found that MKP-1 deficiency caused a considerable increase in K63-linked polyubiquitination on TRAF6 concurrent with a decrease in K48-linked ubiquitination. Increased K63-linked polyubiquitinated TRAF6 reflected on higher levels of activated TAK1 in MKP-1⁻/⁻ BMDMs.

Ubiquitination is critical for a variety of cellular processes (Suresh et al, 2016), including different aspects of immune functions, such immune activation and protein degradation (Reyes-Turcu et al, 2009). Some specific ubiquitination such as K48-linked polyubiquitination targets proteins for degradation via 26S proteasome, whereas the K63-linked polyubiquitination mediates signal transduction and serves as a signaling platform to facilitate protein–protein interactions and propagate signal transduction. Ubiquitination is a reversible and dynamic event that is regulated by various ligases and deubiquitinases. The conjugated ubiquitin chains can be cleaved by a family of USPs or DUBs. Just like phosphatases that counteract the actions of the kinases, DUBs serve as a negative regulator of the ubiquitination process to modulate the actions of the ubiquitination machinery. Our results demonstrate that MKP-1 modulates the E3 ligases (TRAF6) and various USPs, including A20.

The activation of TLR pathway involves phosphorylation and polyubiquitination. Upon IL-1/TLR stimulation, IL-1R1 forms a complex with IL-1RAcP (a co-receptor) and recruits the adapter protein MyD88 to the receptor signaling complex (Cohen & Strickson, 2017). MyD88, in turn recruits the serine/threonine kinases IRAKs (Cohen, 2014). Once IRAKs are recruited to the receptor complex, it is activated and then dissociated from the receptor complex. Dissociated IRAK1 interacts with TRAF6 (Cohen & Strickson, 2017). This interaction triggers dimerization of RING domain of TRAF6 activating its ligase function to catalyze K63-linked polyubiquitination on targeted proteins (Ye et al, 2002; Branigan et al, 2015). Ubc13 (UBE2N), an E2 ubiquitin–conjugating enzyme, is the major E2 assembly catalyzing K63-linked ubiquitin chains, which provides K63-linked polyubiquitin chains to E3 ligases including TRAF6 (Xia et al, 2009). MKP-1–deficient BMDMs exhibited constitutively higher expression of Ubc13 without further increase in response to LPS challenge. As Ubc13 provides TRAF6 the K63 ubiquitin chains to catalyze K63-linked polyubiquitination on targeted proteins (Fukushima et al, 2007), up-regulation of Ubc13 may represent a coregulatory mechanism to supply for the enhanced demand for K63 chains in MKP-1–deficient cells. K63-linked polyubiquitination on TRAF6 is required for the activation of TAK1 and subsequent activation of NFκB and MAPK pathways (Deng et al, 2000; Akira & Takeda, 2004). Surprisingly, we found that MKP-1 deficiency leads to up-regulation of several key IL-1/TLR signaling molecules and this up-regulation was associated with predominant K63 polyubiquitination on TRAF6. To our surprise, A20/TNFAIP3, perhaps the best-known negative regulator of TLR/TRAF6 signaling, was highly abundant in MKP-1$^{-/-}$ macrophages (Fig 4A). The kinetics of A20 induction in response to LPS treatment in WT and MKP-1 BMDMs was similar. Although, MKP-1–deficient BMDMs show much higher basal level, which decreased 30 min after LPS treatment, but rapidly increased after 1 h treatment. The A20 levels peaked in both cell types at 3–6 h post LPS treatment. The enhanced A20 expression in MKP-1–deficient macrophages neither prevented the up-regulation of TRAF6 K63-linked ubiquitination (Fig 3A) nor inhibited TAK1 activation (Fig 2I). We identified that A20 is not only up-regulated in MKP-1–deficient macrophages but also exhibits an increased phosphorylation on 381 serine residues (Fig 4A and C), although we could not exclude phosphorylation on other A20 residues. Previous studies suggested that phosphorylated form of A20 is more effective to cleave the ubiquitin chains and thus inhibits the inflammatory signaling (Hutti et al, 2007; Martens & van Loo, 2019). This finding is surprising and suggests that A20 up-regulation and phosphorylation in MKP-1–deficient macrophages is insufficient to prevent TRAF6 polyubiquitination and activation. Another possibility is that an aberrant phosphorylated form of A20, might not structurally associates with TRAF6 to deubiquitinate the K63 ubiquitin residues. The exact kinase(s) and phosphatase(s) responsible for A20 phosphorylation or dephosphorylation are not precisely defined, but it appears that under inflammatory conditions, NF-κB-activating kinase IKK2 phosphorylates A20 at Ser381 residue (Hutti et al, 2007). Our data suggest that MKP-1 modulates dephosphorylation of A20 either directly or indirectly. If MKP-1 plays a direct role in dephosphorylation of A20, pA20 poises as an unconventional substrate for MKP-1, as MKP-1 dephosphorylate TXY motifs and not Ser residues. Our findings suggest that the hyper-

inflammatory phenotype of MKP-1$^{-/-}$ mice is due to a complex interplay between aberrant phosphorylation and polyubiquitination.

Signaling events rely on tight control and sequential ubiquitination and deubiquitylation as well as phosphorylation and dephosphorylation. Deubiquitinating enzymes (DUBs) are highly selective by cleaving the specific ubiquitin linkage types. Through specific cleavage, DUBs remove the targeted ubiquitin molecules from the modified proteins resulting either in the inactivation of the signaling pathway or rescue of the protein/signaling molecules from degradation (Wilkinson, 2009; Heideker & Wertz, 2015). MKP-1 deficiency led to an imbalance of K63 versus K48 polyubiquitination that was associated with a substantial increase in the expression of a number of deubiquitinases, including USP13 and TNFAIP3 (Fig 1). The role of these USP proteins in the regulation of TRAF6 is intriguing. Pharmacological inhibition of the DUBs with DUBi or Usp13i altered TRAF6 levels and its ubiquitination as well as TAK1 activity in MKP-1$^{-/-}$ macrophages with little effect in WT macrophages (Fig 5). Pharmacological inhibition of USP13 decreased the expression of K63-linked polyubiquitination on TRAF6 and decreased pTAK1 as well as significantly reduced the production of IL-1β and TNF-α (Fig 6B–D), further highlighting the importance of altered deubiquitination in the enhanced inflammatory response of MKP-1–deficient BMDMs. On the other hand, because DUBi is a broad nonspecific pharmacological inhibitor targeting many DUBs, it is not surprising that K48-ubiquitination on TRAF6 was enhanced. It is worth noting that the effects of these inhibitors on WT macrophages appeared to be different from the effects on MKP-1–deficient macrophages (Figs 5A and 6C–F). We speculated that in WT BMDMs these inhibitors are not effective due to lack of aberrant ubiquitination on TRAF6. This observation is intriguing as it may provide a basis for the development of precision drugs. In contrast, MKP-1–deficient macrophages appear to be more sensitive to these inhibitors in respect to TRAF6-mediated TAK1 activation and cytokine production. These data also suggest that the inhibitory effect of Usp13i is precise and specific as it depends upon the differential expression of TLR signaling proteins and the types of their polyubiquitination. Taken together, our findings suggest that MKP-1 deficiency modulates the phosphorylation of many proteins and subsequently impacts on ubiquitination/deubiquitylation processes leading hyperinflammation.

An important finding of this study is the wide disturbance in the enzymes that either ubiquitinate or deubiquitinate proteins in MKP-1–deficient macrophages or mice suggest that MKP-1 deficiency may have a ripple effect on a variety of signaling processes. It is particularly interesting to note that basal levels of IL-1R1, TRAF6, IRAK1, pTAK1 (Fig 2), and A20 (Fig 4) were significantly higher in MKP-1–deficient cells compared with WT cells. These results suggest that MKP-1$^{-/-}$ cells are primed to react to TLR stimulation with a more robust inflammatory response. In summary, we show that MKP-1 deficiency is associated with a maladaptive response in term of TRAF6 polyubiquitination and its negative regulator A20 phosphorylation leading to an enhanced TRAF6 signaling. This evidence suggests that MKP-1 deficiency leads to a disturbance in the expression of many DUBs including A20 leading to increased TRAF6 signaling. These results indicate that in addition to regulate MAP kinase phosphorylation, MKP-1 also exerts a feedback control on the early signaling complexes upstream of the MAP kinase pathway.

# Materials and Methods

## Chemicals and antibodies

All chemicals were purchased from Sigma-Aldrich unless specified otherwise. LPS was purchased from InvivoGen. Usp13 inhibitor (Usp13i) and deubiquitinate enzyme inhibitor (DUBi, PR-619) were purchased from Selleckchem. HRP-conjugated anti-mouse and anti-rabbit IgG secondary antibodies, IRAK1, pTAK1, TAK1, K63 polyubiquitin, and K48 polyubiquitin antibodies were purchased from Cell Signaling Technology. $\beta$-actin, pIRAK1, and pA20 antibodies were purchased from Thermo Fisher Scientific. Antibodies against IL-1R1, TRAF6, Ubc13, A20, and anti-goat IgG secondary antibodies were purchased from Santa Cruz Biotechnology. The IL-1$\beta$ antibody was purchased from R&D Systems. Lambda protein phosphatase (PP) was purchased from New England Biolabs.

## Mice and isolation of bone marrow–derived macrophages

Animal studies were approved by the corresponding University Committee of the Columbus Children's Research Institute on Use and Care of Animals (01505AR, 9 January, 2017). MKP-1$^{-/-}$ mice and the MKP-1$^{+/+}$ (WT) were on a C57/129 mixed background as previously described (Zhao et al, 2005; Frazier et al, 2009; Talwar et al, 2017a). BMDMs from mice were prepared as described previously (Bauerfeld et al, 2012; Talwar et al, 2017a). Briefly, femurs and tibias from 6- to 12-wk-old mice were dissected, and the bone marrow was flushed out. Macrophages were cultured with IMDM media supplemented with 30% L929 supernatant containing glutamine, sodium pyruvate, 10% heat-inactivated FBS, and antibiotics for 5–7 d. BMDMs were re-plated at a density of $2 \times 10^6$ cells/well the day before the experiment. BMDMs from MKP-1$^{+/+}$ (WT) and MKP-1$^{-/-}$ mice were challenged with LPS (100 ng/ml) in the absence or presence of inhibitors (DUBi or USP13i) for different time periods as indicated in each experiment. After treatment, cell lysates were harvested for protein analyses and conditioned media was collected to measure the cytokines.

## Protein extraction and immunoblotting

After the treatments, the cells were harvested and washed twice with PBS. RIPA buffer containing a protease inhibitor and anti-phosphatase cocktail inhibitors II and III (Sigma Chemicals) was added for protein extraction. Proteins (10–25 $\mu$g) were mixed with equal amounts of 2× sample buffer (20% glycerol, 4% sodium dodecyl sulfate, 10% 2-$\beta$ME, 0.05% bromophenol blue, and 1.25 M Tris–HCl, pH 6.8). The proteins were fractionated on a 10% sodium dodecyl sulfate–polyacrylamide gel and transferred to polyvinylidene difluoride (PVDF) membrane using a semi dry Transfer Cell (Bio-Rad), which was run at 18 V for 1 h. The membranes were blocked for 1 h in 5% nonfat dry milk in TBST (Tris-buffered saline with 0.1% Tween 20) and washed. The PVDF membranes were incubated with the primary antibody overnight at 4°C. The blots were washed three times with TBST and incubated for 2 h with the HRP-conjugated secondary anti-IgG antibody (1/5,000 in 5% nonfat dry milk in TBST) at room temperature. After this incubation, the membranes were washed three times in TBST and incubated for 5 min with a chemiluminescent reagent (GE Healthcare) to visualize immunoreactive bands. Images were captured on Hyblot CL film (Denville; Scientific, Inc) using JPI automatic X-ray film processor model JP-33. Optical density analysis of signals was carried out using Image J software.

## Immunoprecipitation

BMDMs were lysed with RIPA buffer (Millipore) containing protease inhibitor and anti-phosphatase cocktails, and the lysate was centrifuged to pellet the cell debris. The resulting supernatant was then used for immunoprecipitation. The lysate was incubated with TRAF6 antibody with 0.1% BSA overnight at 4°C with gentle rotation. Protein A-Sepharose beads (30 $\mu$l) were added and incubated for 2 h at 4°C with gentle rotation. Immunoprecipitation pellets were washed five times using ice-cold wash buffer (50 mM Tris·Cl, pH 7.4, 5 mM EDTA, 300 mM NaCl, 0.1% Triton X-100, and 0.02% sodium azide) with a brief centrifugation each time in a refrigerated microcentrifuge. Samples from the pellets were resuspended in the loading buffer and were subjected to gel electrophoresis and then was blotted onto PVDF membranes (Bio-Rad). Membranes were blocked and incubated with specific primary antibodies. The next day membranes were washed and incubated with HRP-conjugated secondary antibody for 2 h at room temperature and signal was detected using a chemiluminescence reagent as previously described (Talreja & Samavati, 2018).

## ELISA

The levels of IL-1$\beta$ and TNF-$\alpha$ in the conditioned medium were measured by ELISA according to the manufacturer's instructions (ELISA DuoKits; R&D Systems) (Rastogi et al, 2011; Talwar et al, 2017a; Talreja & Samavati, 2018).

## RNA-seq library data analysis

The data analysis was performed on our previously published RNA-seq data obtained from the RNA-seq libraries prepared from frozen liver tissue as described previously (Li et al, 2018; Bauerfeld et al, 2020). The sequencing reads were mapped to the Genome Reference Consortium GRCm38 (mm10) murine genome assembly using TopHat2 (version 2.1.0), and feature counts were generated using HTSeq (version 0.6.1). Statistical analysis for differential expression was performed using the DESeq2 package (version 1.16.1) in R, with the default Benjamini–Hochberg $P$-value adjustment method. Statistically significant differential expression thresholds included an adjusted $P$-value < 0.05 and an absolute value linear fold change of two or greater. Gene Trail 2 analysis tool was used to identify the cellular pathways and gene ontology categories that were over-represented based on the list of DE genes (FDR < 5%). The pathways with FDR < 5% were considered to be significant. The type of FDR adjustment used was Benjamini and Hochberg. Heat maps were generated using the DEseq2 package in R-studio.

## RNA extraction and quantitative reverse transcriptase/real Time-PCR (qRT-PCR)

RNA was isolated from BMDMs and qRT-PCR was performed as described previously (Talwar et al, 2017a, 2017b; Talreja & Samavati, 2018). Relative expression levels were calculated and normalized to GAPDH using $\delta$ delta Ct method. Statistical analysis was performed using paired, two-tailed $t$ test. The following primers were used for PCR reactions: GAPDH forward; 5′-CATCACTGCCACCCAGAAGACTG-3′ and reverse; 5′-ATGCCAGTGAGCTTCCCGTTCAG-3′, TRAF6 forward; 5′-AAA GCG AGA GAT TCT TTC CCT G-3′ and reverse; 5′-ACT GGG GAC AAT TCA CTA GAG C-3′; IL-1R1 forward; 5′-GCA CGC CCA GGA GAA TAT GA-3′ and reverse; 5′-AGA GGA CAC TTG CGA ATA TCA A-3′, TNFAIP3 forward; 5′-AGC AAG TGC AGG AAA GCT GGC T-3′ and reverse; 5′-GCT TTC GCA GAG GCA GTA ACA G-3′, USP13 forward; 5′-GAC CAG AAA GGT TCG CTA CAC G-3′and reverse, 5′-TTC TGC TTC CCT CCG CAT GAG T-3′.

## Statistical analyses

One-way analysis of variance test and post hoc repeated measures comparisons (least significant difference) were performed to identify differences between groups. ELISA results were expressed as mean ± SEM. For all analyses, two-tailed $P$-values of less than 0.05 were considered to be significant.

# Data Availability

All data are available. The gene expression data supporting the findings of this study are available at the https://www.ncbi.nlm.nih.gov/geo/query/acc.cgi?acc=GSE122741.

# Supplementary Information

# Acknowledgements

This work was supported by a grant R01HL150474 (L Samavati), and R01 AI124029 as well as an R21 AI142885 (Y Liu). We thank Bristol Myer Squibb for providing mice.

## Author Contributions

J Talreja: data curation, formal analysis, investigation, methodology, and writing—original draft.
C Bauerfeld: writing—review and editing.
X Wang: methodology.
M Hafner: methodology.
Y Liu: supervision and writing—review and editing.
L Samavati: conceptualization, supervision, funding acquisition, validation, project administration, and writing—review and editing.

## Conflict of Interest Statement

The authors declare that they have no conflict of interest.

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
