## [Reviewer comments · Life Science Alliance]

Life Science Alliance

MKP-1 modulates Ubiquitination Phosphorylation of TLR signaling

Jaya Talreja, Christian Bauerfeld, Xiantao Wang, Markus Hafner, Yusen Liu, and Lobelia Samavati

DOI: <https://doi.org/10.26508/lsa.202101137>

Corresponding author(s): Lobelia Samavati, Wayne State University

Review Timeline:	Submission Date:	2021-06-17
	Editorial Decision:	2021-08-12
	Revision Received:	2021-09-05
	Editorial Decision:	2021-09-08
	Revision Received:	2021-09-13
	Accepted:	2021-09-15

Scientific Editor: Novella Guidi

Transaction Report:

August 12, 2021

Re: Life Science Alliance manuscript #LSA-2021-01137-T

Lobelia Samavati
Wayne State University
Center for Molecular Medicine and Genetics
540 E Canfield St,
Detroit,, Michigan 48201

Dear Dr. Samavati,

Thank you for submitting your manuscript entitled "MKP-1 modulates Ubiquitination/Phosphorylation of TLR signaling" to Life Science Alliance. The manuscript was assessed by expert reviewers, whose comments are appended to this letter. As you will note from the reviewers' comments below, Reviewer 1 is quite excited about these findings and just have few comments on clarifications that need to be addressed in the text, intro and Methods sections. However, Reviewer 2 does have some concerns, and finds that the overall findings only advance the understanding of the phenotype of MKP-1-deficiency marginally and lack any broader impact. Thus, this Reviewer suggests some experiments to be conducted to bring more novelty to this work like showing that reconstitution of MKP-1 KOs with exogenous MPK-1 rescues the enhanced inflammation (increased cytokines) in cells and examining the effects of an IRAK1/4 kinase inhibitor on basal TRAF6 expression in WT vs MKP-1 KO cells. IRAK1 interaction with TRAF6 should be also examined via TRAF6 IP and generally NF- κ B signalling (e.g. I κ Ba degradation and phosphorylation, P-p65, p65 nuclear translocation by IF etc) in the MKP-1 KO cells, would be also a logical downstream TF to examine. We, thus, encourage you to submit a revised version of the manuscript back to LSA that responds to all the reviewers' points.

Thank you for this interesting contribution to Life Science Alliance. We are looking forward to receiving your revised manuscript.

Sincerely,

- A letter addressing the reviewers' comments point by point.
- An editable version of the final text (.DOC or .DOCX) is needed for copyediting (no PDFs).
- High-resolution figure, supplementary figure and video files uploaded as individual files: See our detailed guidelines for preparing your production-ready images, <https://www.life-science-alliance.org/authors>
- Summary blurb (enter in submission system): A short text summarizing in a single sentence the study (max. 200 characters including spaces). This text is used in conjunction with the titles of papers, hence should be informative and complementary to the title and running title. It should describe the context and significance of the findings for a general readership; it should be written in the present tense and refer to the work in the third person. Author names should not be mentioned.

B. MANUSCRIPT ORGANIZATION AND FORMATTING:

Reviewer #3 (Comments to the Authors (Required)):

The present manuscript by Talreja et al entitled "MKP-1 modulates Ubiquitination/Phosphorylation of TLR signaling" demonstrated deficiency of MAPK phosphatase (MKP-1) modulates TRAF-6 activity via K63 ubiquitination. The authors demonstrated IL1R/TLR signaling pathway is disturbed in MKP-1 deficient mice. Authors also showed increased expression of IL-1R1, IRAK1, and TRAF6 in

MKP-1 deficient mice in bone marrow-derived murine macrophages (BMDMs) in MKP-1 deficient mice. The present study is interesting, well written, and fully supported with appropriate references.

Few comments:

1. As manuscript's focus is MKP-1 protein, so the introduction should be rearranged as 1) description of MKP-1, 2) then its link with TLR signaling, and 3) followed by ubiquitination process.
2. MKP-1 deficient mice should be discussed in the materials and method section with appropriate reference.
3. A better description of LPS challenged BMDM is required in the materials and method section.
4. Author should follow the same nomenclature in the manuscript for "MKP-1 deficient mice or MKP-1^{-/-} mice to avoid readers' confusion.
5. On page 14, in the last para, K43 should be replaced by K48.
6. In fig. 4C, the basal phospho A20 is elevated, is there any explanation on why the level is upregulated without any treatment?
7. In Fig. 4B, the 30 mins timepoint showed A20 level less than the basal. Is that due to the reason that A20 is downregulated at first 30 minutes? Although the Western blot image didn't show downregulation.

Reviewer #4 (Comments to the Authors (Required)):

In the manuscript by Talreja et al., the authors examine the function of the phosphatase MKP-1 in innate signalling, beyond its known ability to control prolonged MAPK activation. They demonstrate that loss of MKP-1 leads to increases in signalling molecule expression potentially via dysregulation of DUBs and the E3 ligase response of TRAF6. While the line of investigation is interesting from a mechanistic standpoint the overall findings only advance the understanding of the phenotype of MKP-1-deficiency marginally and lack any broader impact. Hence, this manuscript is likely to only be of interest to those specifically studying MKP-1 or closely related molecules.

Further investigations into other aspects of innate signalling (e.g. impacts on NF- κ B activation) or diving deeper into some of the presented findings should be undertaken to bring more novelty to this work. Some specific comments and suggestions on how to improve the manuscript are outlined below:

1. General: An important experiment would be to show that reconstitution of MKP-1 KOs with exogenous MPK-1 rescues the enhanced inflammation (increased cytokines) in cells? Does it also return some of the new observations to base line as well? e.g. increased TRAF6, DUB etc...
2. Figure 1A: It would be good to show the expression of some other key regulators in these pathways that remained unchanged e.g. IRAK4, MyD88, TRIF?

3. Figure 1B: As above for DUBs that are unchanged.
4. Figure 1C-D, G-H (or in RNAseq data): Although it has previously been shown, it would be good to show the changes in expression of cytokine genes, which should also be increased e.g. TNF, IL-1b, IL-6 and so on.
5. Figure 2A: It appears that there is a much greater increase at the protein level than gene - Does this mean both transcriptional and PTM are involved?
6. Figure 2C: The blots need to be shown in the same experiment on the same membrane - i.e. not cut in the middle. Also, a control that is not induced basally should be shown e.g. IRAK4, IRAK2?
7. Figure 2E: As above - must be shown on same membrane. Also, total IRAK1 should be shown in the same experiment as well as Actin.
8. Given the effect on IRAK1 and TRAF6 it would be of interest to examine the effects of an IRAK1/4 kinase inhibitor on basal TRAF6 expression in WT vs MKP-1 KO cells.
9. Figure 2: To try to bring some more novelty to this manuscript the authors could also examine NF- κ B signalling (e.g. I κ Ba degradation and phosphorylation, P-p65, p65 nuclear translocation by IF etc) in the MKP-1 KO cells, assuming this has not previously been published? Given the increased expression of IRAK1, TRAF6, TAK1 it would be a logical downstream TF to examine, especially given its importance for cytokines that are increased in MKP-1 KOs (e.g. TNF).
10. Figure 3B: Do you see similar increases in TRAF6 if treat WT cells and KOs with MG132 to inhibit proteasomal degradation (i.e. via K48 Ub)?
11. Figure 4B: States combined from 3 experiments but no error bars are shown?
12. Figure 4C: Total A20 should also be shown in this experiment.
13. Figure 5: Could IRAK1 interaction with TRAF6 also be examined via TRAF6 IP here? Is the interaction increased in the KOs?

Dear Dr. Novella Guidi and Editorial Board Member,

Thank you for considering our manuscript for publication in the Life Science Alliance.

We appreciate the thorough review of our manuscript and the efforts put forward by the reviewers and for their excellent and thought-provoking questions that deserve further attention. We also appreciate the positive comments of the reviewers.

We addressed all the points and concerns raised by the editors and reviewers in the revised form of the manuscript. The point-by-point response to address the reviewers' concerns is provided as a separate file labeled "Reply to Referees". We revised the results and discussion according to the reviewers' suggestions.

Reply to Referees

Reviewer #3 (Comments to the Authors (Required)):

The present manuscript by Talreja et al entitled "MKP-1 modulates Ubiquitination/Phosphorylation of TLR signaling" demonstrated deficiency of MAPK phosphatase (MKP-1) modulates TRAF-6 activity via K63 ubiquitination. The authors demonstrated IL1R/TLR signaling pathway is disturbed in MKP-1 deficient mice. Authors also showed increased expression of IL-1R1, IRAK1, and TRAF6 in MKP-1 deficient mice in bone marrow-derived murine macrophages (BMDMs) in MKP-1 deficient mice. The present study is interesting, well written, and fully supported with appropriate references.

Few comments:

1. As manuscript's focus is MKP-1 protein, so the introduction should be rearranged as 1) description of MKP-1, 2) then its link with TLR signaling, and 3) followed by ubiquitination process.

Reply: We agree with the reviewer's suggestion and have rearranged the introduction in the revised manuscript.

2. MKP-1 deficient mice should be discussed in the materials and method section with appropriate reference.

Reply: As suggested we have discussed about the MKP-1 deficient mice in the materials and methods section.

3. A better description of LPS challenged BMDM is required in the materials and method section.

Reply: As suggested by the reviewer we have described the LPS challenge of BMDMs in more details in the materials and methods section.

4. Author should follow the same nomenclature in the manuscript for "MKP-1 deficient mice or MKP-1^{-/-} mice to avoid readers' confusion.

Reply: According to reviewer's suggestions, we have changed the nomenclature for "MKP-1 deficient mice to MKP-1^{-/-}.

5. On page 14, in the last para, K43 should be replaced by K48.

Reply: We are thankful to the reviewer for pointing out the mistake. We corrected it.

6. In fig. 4C, the basal phospho A20 is elevated, is there any explanation on why the level is upregulated without any treatment?

Reply: This is a great point. Both unphosphorylated and phosphorylated A20 is increased in MKP-1 deficiency. This observation led us to further investigate A20 in MKP-1 deficient cells. As shown, at the basal levels (without LPS or E. Coli stimulation) A20 protein and mRNA levels are elevated in MKP-1 deficiency. Our thought process is that because TRAF6 is highly ubiquitinated, it leads to A20 upregulation as a negative feedback mechanism. As we indicated in our MS, pathway analysis in MKP-1 deficiency showed TLR and ubiquitination pathways are highly impacted in MKP-1^{-/-} mice and cells. Previously, we have shown that MKP-1 deficient cells exhibit elevated levels of pp38 at baseline (Cell Signal. 2017 Jun; 34:1-10. doi: 10.1016/j.cellsig.2017.02.018. Epub 2017 Feb 24, PMID: 28238855) and this may result in elevated phosphorylated A20.

7. In Fig. 4B, the 30 mins timepoint showed A20 level less than the basal. Is that due to the reason that A20 is downregulated at first 30 minutes? Although the Western blot image didn't show downregulation.

Reply: We agree with the reviewers. Previous studies have shown that A20 is induced upon LPS activation within 1-4h. Most other studies show only one time point in the Western blot for a given protein. Here, we diligently provided a time course for most of the signaling molecules we studied. First, the A20 upregulation pattern in WT and MKP-1 deficient cells are similar. Secondly, both WT and MKP-1^{-/-} BMDMs show an early dip in A20 in response to LPS. It appears that in MKP-1 deficient BMDMs the decrease is more pronounced. This is because the baseline is much higher in MKP-1^{-/-} cells as compared to MKP-1^{+/+} cells. Previously, we have observed this phenomenon (see J Biol Chem. 2013 Nov 22;288(47):33966-77. doi: 10.1074/jbc.M113.492702), various mechanisms may be responsible for this observation, including rapid degradation of proteins or delayed translation. However, our focus in this paper was not the A20, further studies need to determine the mechanisms of A20 regulation.

Reviewer #4 (Comments to the Authors (Required)):

In the manuscript by Talreja et al., the authors examine the function of the phosphatase MKP-1 in innate signalling, beyond its known ability to control prolonged MAPK activation. They demonstrate that loss of MKP-1 leads to increases in signalling molecule expression potentially via dysregulation of DUBs and the E3 ligase response of TRAF6. While the line of investigation is interesting from a mechanistic standpoint the overall findings only advance the understanding of the phenotype of MKP-1-deficiency marginally and lack any broader impact. Hence, this

manuscript is likely to only be of interest to those specifically studying MKP-1 or closely related molecules.

Further investigations into other aspects of innate signalling (e.g. impacts on NF- κ B activation) or diving deeper into some of the presented findings should be undertaken to bring more novelty to this work. Some specific comments and suggestions on how to improve the manuscript are outlined below:

1. General: An important experiment would be to show that reconstitution of MKP-1 KO cells with exogenous MPK-1 rescues the enhanced inflammation (increased cytokines) in cells? Does it also return some of the new observations to base line as well? e.g. increased TRAF6, DUB etc...

Reply: The reviewer's suggestion about reconstitution of MKP-1 KO cells with exogenous MKP-1 sounds good, however, we chose as comparison WT (MKP-1^{+/+}) mice which are expressing sufficiently MKP-1 protein and are from the same background as of MKP-1^{-/-} mice. Overexpression of MKP-1 requires the use of adenoviruses expressing MKP-1. The infection with viral particle has a tremendous effect on cell signaling especially on MyD88 dependent pathways, including, TLRs, IRAKs and TRAF6. Viral infection stimulates immune responses and may have different effect on ubiquitination. Secondly, macrophages are terminally differentiated cells and it is difficult to achieve higher levels of transfection and meaningful upregulation of MKP-1 in MKP-1 deficient BMDMs.

2. Figure 1A: It would be good to show the expression of some other key regulators in these pathways that remained unchanged e.g. IRAK4, MyD88, TRIF?

Reply: Thank you for this comment. We found a large number of genes (more than 20% of entire murine genome) differentially expressed in MKP-1 KO mice (Life Sci. 2020 Jan 15; 241: 117157. Published online 2019 Dec 16. doi: 10.1016/j.lfs.2019.117157.

The results are published in public domain and data are readily available to the research community. (<https://www.ncbi.nlm.nih.gov/geo/query/acc.cgi?acc=GSE122741>).

As suggested by reviewer the graphs below show the expression of some other key regulators in the TLR pathway that were unchanged at the gene level. It shows that there is no difference in the expression of IRAK4 at basal levels as well as in response to *E. coli* infection between WT and MKP-1 KO macrophages (A). Similarly, there was no difference in the basal expression of MyD88 (B) and TRIF (C) between WT and MKP-1 KO mice. However, in response to *E. coli*, MyD88 was higher expressed in MKP-1 KO.

Based on reviewer suggestion, we performed new experiment and now provide in the new Figure 2C, showing IRAK 4 protein level is increased in MKP-1^{-/-} as compared to MKP-1^{+/+} cells (WT), although the IRAK4 at the gene level was not significantly altered between MKP-1^{-/-} and MKP-1^{+/+} (see below). These results indicate an altered protein degradation

in MKP-1^{-/-}. This is the whole reseau we investigated the ubiquitination pathway in this manuscript.

We have included the above Figure in Revised Supplementary Figure S1(A-C) and written about the figure in the results section of the manuscript text.

3. Figure 1B: As above for DUBs that are unchanged.

Reply: As suggested by reviewer the graphs below show the expression of few DUBs that were unchanged. The results show that there were no differences in the expression levels of CYLD and USP21 between WT and MKP-1 KO mice.

We have included the above Figure in Revised Supplementary Figure S1(D&E) and written about the figure in the results section of the manuscript text.

4. Figure 1C-D, G-H (or in RNAseq data): Although it has previously been shown, it would be good to show the changes in expression of cytokine genes, which should also be increased e.g. TNF, IL-1b, IL-6 and so on.

Reply: Thank you for your comment. As we indicated previously, we found a large number of genes differentially expressed in MKP-1 KO mice.

However, as suggested by reviewer, we reanalyzed the data in this regard. The graphs below show the changes in the expression of cytokine genes. The results show that the expression of IL-1 β , IL-6 and TNF- α markedly increased after E. coli infection in both WT and MKP-1 KO macrophages. There was a significantly higher increase in the expression of IL-1 β and IL-6 in MKP-1 KO macrophages as compared to WT macrophages.

We have included the above Figure in Revised Supplementary Figure S1(F-H) and written about the figure in the results section of the manuscript text.

5. Figure 2A: It appears that there is a much greater increase at the protein level than gene - Does this mean both transcriptional and PTM are involved?

Reply: We agree with the reviewer's observation that there is a much greater increase of IL-1R1 at the protein level than gene. Additionally, we agree that it means both transcriptional and PTM are involved in this process. Previously, we have shown that p38 activation has a huge effect on mRNA translation. Additionally, the current study indicates that ubiquitination may play a role in PTM and protein stability. However, this requires further study.

6. Figure 2C: The blots need to be shown in the same experiment on the same membrane - i.e. not cut in the middle. Also, a control that is not induced basally should be shown e.g. IRAK4, IRAK2?

Reply: The Figure is from the same membrane both the WT and MKP-1KO cell lysates were run on the same blot as shown below. In addition to total cell lysates the cytosolic and nuclear extract lysates were also run on this gel. The Figure in the manuscript shows the

wells 1,2 for WT and wells 7,8 for MKP-1KO. The description of samples in the wells is: 1-6 WT samples and 7-12 MKP-1KO samples. Well 1- whole cell lysates (Untreated Control, well 2 -LPS treated, well 3 (Control) and well 4 (LPS) are cytosolic extracts, well 5 (Control) and well 6 (LPS) are nuclear extracts. Similarly, the samples with the same order of treatment are from 7-12 for MKP-1. In order not to confuse reader, in Fig. 2C, we are showing the basal levels of IRAK1 and changes with LPS treatment in whole cell lysates.

We evaluated the IRAK4 level using specific antibody against IRAK4. As shown in new Fig. 2C, IRAK4 levels are also increased in MKP-1 deficiency. These data indicate that many upstream TLR signaling proteins are increased in MKP-1 KO cells. This suggest that aberrant protein degradation may play a role in MKP-1 KO cells. All these cumulative data led us to investigate polyubiquitination process in MKP-1 KO cells.

As suggested by reviewer we reblot the same blot with the IRAK4 antibody as a control and have included in revised Fig.2C.

7. Figure 2E: As above - must be shown on same membrane. Also, total IRAK1 should be shown in the same experiment as well as Actin.

Reply: Thank you for your comment. In fact, the Figure is from the same membrane i.e both the WT and MKP-1KO cell lysate samples were run on the same gel as shown below treated with LPS for different time periods: 0, 15 min, 30 min, 1h, 3h and 6h both in WT and MKP-1KO macrophages. The reason we cut in the middle of gel, we wanted to emphasize the early differences between WT and MKP1 knockout mice. Additionally, at later time points we did not see a significant difference between the mice strains.

8. Given the effect on IRAK1 and TRAF6 it would be of interest to examine the effects of an IRAK1/4 kinase inhibitor on basal TRAF6 expression in WT vs MKP-1 KO cells.

Reply: We agree with reviewer' suggestion whole heartedly. However, we provided a lot of data in this manuscript and the interpretation is already complicated. We think this is an important subject and it is better to be addressed in a separate study.

Many years back we identified that in sarcoidosis, a human disease associated with inflammatory granulomatous disease, the macrophages exhibit a lack of MKP-1 expression. We found that sarcoidosis patients exhibit MKP-1 deficiency and have elevated levels of pp38. In fact, sarcoidosis patients have a lot of similarities with MKP-1 knockout mice. Using alveolar macrophages of sarcoidosis patients, we used IRAK1/ IRAK 4 inhibitor to investigate the effect on pp38 as well as cytokines. Our results show that IRAK-1/4 inhibitor does not inhibit IL-1 β , IL-6 production or p38 phosphorylation in sarcoidosis AMs and PBMCs (J Immunol. 2016 Aug 15; 197(4): 1368–1378, doi: 10.4049/jimmunol.1600258).

Although it would be very interesting to see if IRAK1/IRAK4 has the same effect in MKP1 knockout, again we believe that this needs to be address in a different study.

9. Figure 2: To try to bring some more novelty to this manuscript the authors could also examine NF- κ B signalling (e.g. I κ B α degradation and phosphorylation, P-p65, p65 nuclear translocation by IF etc) in the MKP-1 KO cells, assuming this has not previously been published? Given the increased expression of IRAK1, TRAF6, TAK1 it would be a logical downstream TF to examine, especially given it importance for cytokines that are increased in MKP-1 KOs (e.g. TNF).

Reply: The reviewer has raised a very interesting question. However, previously, we have addressed this issue and results are published. These results show that activation of IKKa/b occurred within 15 min after LPS stimulation, then IKKa/b phosphorylation rapidly declined. No obvious difference in IKKa/b activity was observed between WT and MKP-1 KO BMDMs, although I κ B levels appeared to recover faster in MKP-1KO than in WT BMDMs (J Immunol 2021; 206:2966; doi:10.4049/jimmunol.2001468). We have also shown that there are no differences in NF- κ B DNA binding ability between WT and MKP-KO macrophages either before or after LPS stimulation (JBC, VOL. 284, NO. 40, pp. 27123–27134, October 2, 2009).

10. Figure 3B: Do you see similar increases in TRAF6 if treat WT cells and KOs with MG132 to inhibit proteasomal degradation (i.e. via K48 Ub)?

Reply: We agree with the reviewer's suggestion. We think it is better to address this issue in a separate study. However, MG132 is a non-specific proteasomal inhibitor that reduces the degradation of K48-ubiquitin-conjugated proteins in cells. Our results show that MKP-1 KO cells exhibit a significantly higher level of K63-linked polyubiquitinated TRAF6 and lower levels of K48-linked polyubiquitinated TRAF6 as compared to WT BMDMs both at baseline and after LPS stimulation. Therefore, inhibition of proteasomal degradation with MG132 might not have any effect on the levels of TRAF6. We think enhanced K63 ubiquitination of MKP-1 is the upstream effect before targeting to proteasome.

11. Figure 4B: States combined from 3 experiments but no error bars are shown?

Reply: We apologize for not showing the error bars by mistake. We have revised the Figure 4B that shows error bars.

12. Figure 4C: Total A20 should also be shown in this experiment.

Reply: We agree that the phosphorylated proteins and the total protein should be shown in the figures. Yet, A20 is an inducible protein and we have shown in the Figure 4A, how total A20 is induced in WT and MKP-1 KO BMDMs. The figure shows that the MKP-1 KO cells exhibit higher levels of A20 at baseline and in response to LPS activation as compared to WT. However, to comply with reviewer suggestion, we blotted with total A20 as shown below. We have revised Fig.4C as suggested.

13. Figure 5: Could IRAK1 interaction with TRAF6 also be examined via TRAF6 IP here? Is the interaction increased in the KOs?

Reply: We are thankful to the reviewer for this suggestion. As suggested by the reviewer we performed new experiments and examined the interaction between TRAF6, IRAK1 after TRAF6 IP. The result shows an increased interaction between IRAK1 and TRAF6. We have included this new blot as Figure 3C in revised figures.

Overall, we thank both reviewers for their excellent suggestions, which improved the clarity of our manuscript.

September 8, 2021

RE: Life Science Alliance Manuscript #LSA-2021-01137-TR

Prof. Lobelia Samavati
Wayne State University
Center for Molecular Medicine and Genetics
540 E Canfield St,
Detroit,, Michigan 48201

Dear Dr. Samavati,

Thank you for submitting your revised manuscript entitled "MKP-1 modulates Ubiquitination/Phosphorylation of TLR signaling". We would be happy to publish your paper in Life Science Alliance pending final revisions necessary to meet our formatting guidelines.

- please upload your supplementary figures as single files, as well
- please add your main, supplementary figure legends to the main manuscript text after the references section
- please add ORCID ID for the corresponding author-you should have received instructions on how to do so
- please add a Category for your manuscript in our system
- please add the Twitter handle of your host institute/organization as well as your own or/and one of the authors in our system
- please consult our manuscript preparation guidelines <https://www.life-science-alliance.org/manuscript-prep> and make sure your manuscript sections are in the correct order
- please revise a legend for Figure S1 so it matches the figure

Figure check:

- please provide molecular weights alongside each figure with protein blot
- blots in Figure 5E are blurry. If possible, please provide an image that is less exposed
- please add total IRAK1 to blot in figure 2E as already suggested by reviewer 4 and submit source data of this figure as separate file (not only in point by point letter)
- please submit source data as separate file also for figure 2C (not only in point by point)

LSA now encourages authors to provide a 30-60 second video where the study is briefly explained. We will use these videos on social media to promote the published paper and the presenting author. Corresponding or first-authors are welcome to submit the video. Please submit only one video per manuscript. The video can be emailed to contact@life-science-alliance.org

To upload the final version of your manuscript, please log in to your account:
<https://lsa.msubmit.net/cgi-bin/main.plex>

A. FINAL FILES:

B. MANUSCRIPT ORGANIZATION AND FORMATTING:

Thank you for your attention to these final processing requirements. Please revise and format the

manuscript and upload materials within 7 days.

Sincerely,

September 15, 2021

RE: Life Science Alliance Manuscript #LSA-2021-01137-TRR

Prof. Lobelia Samavati
Wayne State University
Center for Molecular Medicine and Genetics
540 E Canfield St,
Detroit,, Michigan 48201

Dear Dr. Samavati,

Thank you for submitting your Research Article entitled "MKP-1 modulates Ubiquitination/Phosphorylation of TLR signaling". It is a pleasure to let you know that your manuscript is now accepted for publication in Life Science Alliance. Congratulations on this interesting work.

*****IMPORTANT:** If you will be unreachable at any time, please provide us with the email address of an alternate author. Failure to respond to routine queries may lead to unavoidable delays in publication.*******

DISTRIBUTION OF MATERIALS:

Again, congratulations on a very nice paper. I hope you found the review process to be constructive and are pleased with how the manuscript was handled editorially. We look forward to future exciting submissions from your lab.

Sincerely,
